# Knowledge, attitudes and practices regarding neonatal jaundice among caregivers in a tertiary health facility in Ghana

Solomon Mohammed Salia[1]*, Agani Afaya[1,2], Abubakari Wuni[3], Martin Amogre Ayanore[4], Emmanuel Salia[5], Doreen Dzidzor Kporvi[1], Peter Adatara[1], Vida Nyagre Yakong[6], Sean Augustine Eduah-Quansah[1], Shine Seyram Quarshie[1], Eric Kwame Dey[1], Dominic Amoah Akolga[1], Robert Kaba Alhassan[7]

1 Department of Nursing, School of Nursing and Midwifery, University of Health and Allied Sciences, Ho, Ghana, 2 College of Nursing, Yonsei University, Seoul, South Korea, 3 Nurses' and Midwives' Training College, Tamale, Ghana, 4 Department of Health Policy Planning and Management, School of Public Health, University of Health and Allied Sciences, Ho, Ghana, 5 Central Laboratory, Korle-Bu Teaching Hospital, Accra, Ghana, 6 Department of Midwifery, School of Nursing and Midwifery, University for Development Studies, Tamale, Ghana, 7 Centre for Health Policy and Implementation Research, Institute of Health Research, University of Health and Allied Sciences, Ho, Ghana

* ssmohammed@uhas.edu.gh

**Data Availability Statement:** All relevant data are within the paper and its Supporting Information files.

## Abstract

### Background

Neonatal jaundice is a major reason babies are frequently re-admitted after hospital discharge following delivery. One means of improving neonatal care and reducing potential mortality associated with neonatal jaundice in resource-limited settings is to create awareness among caregivers. Caregivers who tend to have higher knowledge and awareness, also have positive attitudes, and are not guided by outmoded socio-cultural beliefs and practices are more likely to seek early care and treatment for neonatal jaundice.

### Objective

This study investigated caregivers' knowledge, attitude and practices regarding neonatal jaundice in a tertiary health facility in the Volta region of Ghana.

### Methods

This was a descriptive cross-sectional study that employed a quantitative approach for data collection. A total of 202 caregivers from the Ho Teaching Hospital in the Volta region of Ghana were sampled using a systematic random sampling strategy where quantitative data was collected using a questionnaire and analyzed with STATA version 14.0. Ordered logistic regression was used to determine the factors that were associated with caregivers' knowledge regarding neonatal jaundice and attitude after controlling for relevant covariates.

**Funding:** The author(s) received no specific funding for this work.

**Competing interests:** The authors have declared that no competing interests exist.

**Abbreviations:** ANC, Antenatal Care; REC, Research Ethics Committee; SDGs, Sustainable Development Goals; UHAS, University of Health Allied Science; VIF, Variance Inflation Factor; LMICs, Low- and Middle-Income Countries; GHS, Ghana Health Service.

## Results

Less than half of the caregivers demonstrated good knowledge (45.5%) and attitude (47.5%) but 58.9% had good practices regarding neonatal jaundice. Caregivers who had prior awareness and education on neonatal jaundice were three times more likely to have good knowledge about jaundice than those without previous education [AOR = 3.02, (95% CI: 1.59–5.74), p = 0.001]. A caregiver employed in the public sector was two times more likely to have a good attitude about jaundice than those employed in the private sector [AOR = 2.08, (95%CI: 1.03–4.21), p = 0.042].

## Conclusion

Less than two thirds of the caregivers demonstrated good practice with limited knowledge and poor attitude. Efforts to promote well informed and improved caregivers' attitude will advance positive maternal health-seeking behavior and reduce disabilities and death through early detection and intervention of infants with neonatal jaundice. Public awareness and education about neonatal jaundice especially among caregivers in the private sector should also be intensified.

## Introduction

Empirical evidence indicates that neonatal jaundice is a major reason babies are frequently re-admitted after hospital discharge, and it affects about 60% of term infants and 80% of preterm within the first week of life [1]. The term neonatal jaundice is used to described the yellowish coloration of the skin and other membranes of the newborn, which signifies elevated levels of unconjugated bilirubin in their blood [1, 2]. Blackburn [3] asserts that jaundice results from an imbalance between the rate of bilirubin production and bilirubin excretion leading to increased levels of bilirubin in the blood of the newborn. The presence of jaundice on clinical examination indicates hyperbilirubinemia, which is defined as a total amount of serum bilirubin more than 1.5 mg/dL [4].

Physiological jaundice which is a form of jaundice is the elevation of unconjugated bilirubin in the blood of the newborn occurring during the third to fourth day of life, as a result of an inability of the newborn's liver due to immaturity to convert the unconjugated bilirubin for excretion [5]. It may be benign and self-limiting [6] and may resolve by the end of the first week of life [7]. Factors responsible for the development of this jaundice includes shortened lifespan of red blood cells (70 to 90 days), high number of circulating erythrocytes, lower plasma binding capacity, and delayed passage of meconium [6]. Furthermore, pathological jaundice is the manifestation of jaundice in the newborn within the first 24 hours of life when the serum bilirubin levels rise to more than 5mg/dL and may be due to factors such as ABO and Rh incompatibilities, polycythemia, and septicemia [6].

Although neonatal jaundice affects several babies, many of them recover. In some babies, high levels of unconjugated bilirubin may result in acute and chronic bilirubin encephalopathy or kernicterus leading to irreversible brain damage [8] which may cause death [9, 10]. If jaundice is not detected and treated early in life, it may cause major disabilities in babies such as cerebral palsy, mental retardation, and deafness [11].

Evidence from the Ghana Health Service (GHS) shows that the incidence of neonatal jaundice among newborns has been on the rise in recent years. For instance, from 2015 to 2019,

Ghana recorded 3,031, 4,251, 5,338, 7,175 and 9,273 cases of neonatal jaundice respectively [12]. Also, a study by Adoba et al [13] in Ghana that focused on the knowledge and determinants of neonatal jaundice reported that the prevalence of neonatal jaundice was 66.7%. Nevertheless, neonatal jaundice remains one of the most important contributors to neonatal mortality [14, 15] and as a result, all necessary efforts to reduce its prevalence are imperative.

Ghana's attempt to meet the global targets of reducing neonatal mortality saw the country endorse the health components of the Sustainable Development Goals (SDGs) and the World Health Organization (WHO) targets of reducing neonatal mortality rate to 12 and 7 per 1000 live births by 2030 and 2035 respectively [16–18]. In resource-limited settings, one important measure of reducing neonatal mortality is to create awareness in a bid to improve the knowledge and attitudes of caregivers. This will help dispel some myths and socio-cultural beliefs and misconceptions regarding neonatal jaundice among caregivers who are essential in meeting a continuum of care for neonates after birth.

There is a plethora of data regarding caregivers' perspectives of neonatal jaundice globally [8, 19–25], however, there is a paucity of information regarding caregivers' perspectives of neonatal jaundice in the Ghanaian setting especially, the Volta region. Several studies have reported gaps about caregivers' knowledge and attitude of neonatal jaundice [5, 8, 19, 20, 25–27] in low and middle-income countries including Ghana. In the studies of [22, 25, 27], though some of the mothers of neonates knew the definition and how to recognize jaundice in their babies when it develops, the majority demonstrated knowledge gaps in the causes and treatment. For instance, a majority of the mothers indicated that they did not know the causes and treatment of neonatal jaundice and some mothers however prefer to use herbal treatment and exposure to sunlight as a means of treatment. Furthermore, on the aspect of attitude, the studies of [19, 20, 28] have all reported poor attitude with regards to treatment because mothers delay in seeking medical care when their babies develop jaundice. They will further expose babies to sunlight as means of treatment. It is worth noting that inadequate knowledge and poor practices passed on to parents from previous generations as well as perceptions and attitudes of parents toward neonatal jaundice may explain reasons for delay in seeking medical advice immediately [29] and adherence to inappropriate treatment methods.

To address the above knowledge gaps concerning neonatal jaundice, this study assessed a broad range of caregivers' knowledge, attitudes and practices regarding neonatal jaundice in a tertiary referral facility in the Volta region of Ghana. Understanding the knowledge level, attitude and identifying the practices of caregivers are key in improving newborn survival and reducing neonatal mortality rates. The study findings will also add to the body of knowledge of existing data on neonatal jaundice in Ghana and also serve as useful information for educating expectant parents on appropriate health-seeking behaviors in a bid to reduce neonatal mortality rates in Ghana.

## Materials and methods

### Ethics approval and consent to participate

Ethical approval was obtained from the University of Health and Allied Sciences Research Ethics Committee (UHAS-REC) with protocol number (UHAS-REC/A.7[11]18–19). Administrative approval was also obtained from the management of Ho Teaching Hospital and the nurse in-charge of the postnatal unit. A voluntary written informed consent was obtained from the caregivers which included a well explained description and objectives of the study. Caregivers were informed that they had the right to voluntarily withdraw from the study without any penalty. They were also informed and assured of confidentiality and anonymity during the study by protecting their information and identity. Caregivers were informed that they will not receive direct material benefits for participation, however, the findings of this study will be

published for wider reading and will also help to improve the caring practices of newborns among caregivers and healthcare providers regarding the prevention of neonatal jaundice. They were however informed that their participation in this study will have no risk or discomfort. The caregivers were informed that data collection will be done on an individual basis and that privacy will be provided at all times during the process of data collection.

## Study area

The study was conducted in the Ho Teaching Hospital, Volta region of Ghana. The Ho Teaching Hospital is located in the Volta regional capital (Ho), and the sole tertiary and referral facility in the region. The referral facility is a 313-bed capacity. The hospital has a neonatal intensive care unit (NICU) that provides specialized care to sick babies. The NICU has an admission capacity of 30 beds, six (6) functional incubators and two (2) phototherapy machines. In 2019, the total number of deliveries for the hospital stood at spontaneous vagina deliveries (1006) and caesarian section (803). The NICU recorded 411 admissions in 2019 with a neonatal mortality rate of 11.7% (42 deaths). It also has two pediatricians, three medical officers, 20 professional and four nonprofessional nurses. The postnatal unit operates under the Obstetrics and Gynecology department from Monday to Friday each week. The unit has one doctor and two midwives. Care provided in the unit include general head to toe examination of postnatal clients, neonates, removal of caesarian section stitches and wound care. Additionally, the midwives also educate women on exclusive breastfeeding and perineal hygiene as well as cord care. The hospital provides both general and specialized medical and surgical services including obstetric and gynecological, Ear Nose and Throat, Eyecare among others. Data collection took place at the postnatal unit of the hospital which offers services to parents and their babies after delivery.

## Study design

This was a hospital-based descriptive cross-sectional study that employed a quantitative approach to investigate caregiver's knowledge, attitude and practices regarding neonatal jaundice. This design was adopted because it is good at collecting information that describes a phenomenon as it exists.

## Sample size and sampling determination

The sample size was determined using Yamane [30] formula for sample calculation based on a known population and a total sample of 202 caregivers with babies was determined. The study employed a systematic random sampling strategy for data collection. This method was chosen to ensure that all the caregivers who formed part of the sampling frame had equal chances of being selected to be part of the study. Nurses and midwives gave the caregivers numbers based on the time of arrival at the postnatal unit. The first person was given the number 1 in that order until the last person. Therefore, based on these numbers, the caregivers were arranged to sit in rows. Their names were then documented into a book which served as the register of attendance. In this study, the sampling interval for systematic sampling was determined by dividing the sample size (202) by the total target population of interest (410). The sampling interval was 2. Therefore, every second caregiver in the row was chosen to participate in the study. **Balloting was done** to select the first two caregivers to determine the starting point.

## Study population

The study population were caregivers (father or mother) of the babies who sought care at the postnatal unit of the referral hospital.

### Inclusion/Exclusion criteria

Caregivers aged 18 years and above who sought postnatal services at Ho Teaching Hospital with their babies and consented to participate were included in the study. Those who were present during the period of the study but did not offer voluntary consent were excluded from the study. The study also excluded other caregivers who were not the parents of the babies but brought the babies for postnatal care. Caregivers were recruited for the study for a period of two months from March to April 2019.

### Study variables

The main variables in this study were the Independent and Dependent variables. The independent variables of interest were classified as follows; Age (18–25, 26–35, 36–45 and >45); Sex (Male and Female); Religion (Christian, Muslim and Traditionalist); Marital status (Single, Married, Divorced and Widowed); Ethnicity (Ewe, Akan, Ga and Others); Educational level (None, Basic, Secondary and Tertiary); Residence (Rural, Urban and Estate); Number of previous children (None, One, Two and more than two); Employment status (Self/Private, Public Servant, Unemployed/Student and Unknown); Previous education on neonatal jaundice (Yes or No); Child previously diagnosed with neonatal jaundice (Yes or No); Number of Antenatal Care (ANC) (less than 4 visits and more than 4 visits). Although the WHO has changed the minimum recommended antenatal visits from 4 to 8 [31], in this study, we studied an attendance of 4 visits [32].

The dependent variables of interest for the ordered logistic regression were the composite variables for the overall level of knowledge, attitude and practice. The overall level of knowledge was computed using 28 items grouped to (definition, causes, complications, danger signs, sites for checking, treatment and prevention of neonatal jaundice) that measured knowledge, while nine items each were used to compute for the attitude and practices. These are captured in more detail in the results section.

### Data collection instrument

A structured questionnaire was used to collect information regarding caregivers' knowledge, attitude and practice on neonatal jaundice. The questionnaire designed was informed by the objectives of the study and after a careful review of literature relevant to the study objectives. The instrument was organized into four parts; part one collected information on caregivers' sociodemographic characteristics such age, marital status, educational background, ethnicity, number of children, number of Antenatal Care (ANC) attendance etc. The second part consisted of 28 items that asked knowledge questions in areas of definition, causes, clinical manifestations, danger signs, treatment, complications and prevention. In part three, nine items gathered information on attitude while part four comprised of questions relating to beliefs and practices of neonatal jaundice with 8 items. The questionnaire was ranked on a five-point Likert scale with appropriate descriptions as: 1 = "Strongly disagree", 2 = "Disagree", 3 = "Undecided", 4 = "Agree" and 5 = "Strongly disagree". Questionnaires were serially numbered to allow for easy identification and accuracy of input into data entry sheet for easy analysis.

### Data collection procedure

Data was collected from 1st March, 2019 to 30th April, 2019 at the postnatal unit of the Ho Teaching Hospital. The objectives of the study were explained to the caregivers and voluntarily informed consent was obtained. Items in the questionnaire were explained to the understanding of the caregivers before they were administered. Caregivers were approached at the

postnatal unit on each day of data collection and data collection was done on an individual basis in a private restroom in the unit to ensure privacy. Respondents who were proficient in speaking and writing in the English language answered the questionnaires by themselves. Those who could neither read nor write in the English language were guided to complete the questionnaire. Four of the authors were native speakers of the "Ewe" language and three authors who could speak "Ewe", "Twi" and other languages were used to orally translate the questionnaire into these local languages for the understanding of the caregivers.

Clarity was however given to those who wanted further explanation about the questions. Each day after data gathering, the filled questionnaires were kept in a sealed A4 brown envelope and kept in a safe place from the reach of others to maintain anonymity. All questionnaires were crossed checked for completeness before leaving the data collection site each day.

## Validity and reliability

The questionnaire was peer-reviewed by all authors. Also, two midwives specialized in neonatal nursing and three pediatricians reviewed the questionnaire for content validity and appropriateness of questions. Further validation was done through piloting as well as pretesting which took place in a government hospital in the Ho municipality. The instrument was tested for scale reliability which yielded the following scale reliability coefficients; knowledge (0.78), attitude (0.51) and beliefs and practices (0.84). The reliability coefficient (r) for knowledge (0.78) was measured as adequate, attitude (0.51) was below the required coefficient value of 0.7, and beliefs and practices (0.84) showed a good coefficient value. The closer the reliability coefficient is to 1.0, the greater the internal consistency of the items in the scale. The items in the study instrument showed a good internal consistency except for attitude which showed poor internal consistency. George and Mallery [33] provide a rule of thumb that when the reliability coefficient is below 0.5 it is unacceptable. However, the attitude subscale was not below 0.5 so it was not excluded from the instrument.

## Data analysis

For data quality checks, the field data were first entered into Microsoft Excel and cleaned before exporting to STATA version 14.0 for analysis. Descriptive statistics was used for the demographic characteristics and categorical variables using simple frequencies and percentages.

Overall level of knowledge, attitude and practice were measured by scoring responses that measured caregivers' knowledge, attitude and practices. These were classified as good, moderate and poor. A score of '1' was assigned if respondents had a correct response in knowledge, attitude and practice and '0' if respondents had a wrong response in knowledge, attitude and practice. Total scores were computed and a cumulative score of 80% and above was considered to be good, 60–80% was considered as moderate and below 60% was considered to be poor. The level of knowledge, attitude and practice were coded 0 for poor, 1 for moderate and 2 for good.

Ordered logistic regression analysis was used to determine the association between dependent (knowledge, attitude and practice) and independent variables (socio-demographic characteristics) and statistical significance was considered based on the p-value $<0.05$ at a confidence level of 95%. All explanatory variables of interest were tested for multicollinearity and none had a Variance Inflation Factor (VIF) above 10 necessary for exclusion from the regression model.

## Results

### Socio-demographic characteristics of caregivers

Of the 202 caregivers who participated in the study, a majority of them were aged 26–35 years (56.9%) and 64.9% were married, 31% had two children. Most of them (45.0%) had tertiary education and 35.7% were engaged in self or private employment. A majority, 71.8% attended ANC visit at least four times. Socio-demographic characteristics are presented in Table 1.

**Summary of knowledge, attitude and practice scores of caregivers on neonatal jaundice.** A summary of knowledge, attitude and practice scores of caregivers is presented in Fig 1. The results revealed that a little above 50% of the caregivers (54.5%) and (52.0%) had poor knowledge and attitude regarding neonatal jaundice. However, a majority (58.9%) recorded good practice scores towards neonatal jaundice.

### Caregivers' knowledge on neonatal jaundice

The majority, (89.6%) correctly defined jaundice as the yellowish discoloration of the skin and eyes. About 53% identified prematurity as a cause of neonatal jaundice while 77.7% identified the death of a baby as a complication of jaundice. The majority (91.6%) correctly identified that skin and eyes are the commonest sites for checking jaundice. On the aspect of treatment, less than 50% correctly identified traditional methods as an inappropriate choice of treatment for jaundice (32.7%). Table 2 shows caregivers' knowledge in neonatal jaundice.

### Attitude of caregivers towards neonatal jaundice

The majority of the caregivers 86.1% indicated that jaundice is treated in the hospital in early life, 81.2% correctly indicated that frequent ANC attendance will prevent jaundice in newborns. Furthermore, only 21.8% and 30.2% of the caregivers correctly identified the use of herbal medications and exposing a baby to sunlight as inappropriate modes of treatment of neonatal jaundice. Caregivers attitude towards neonatal jaundice is presented in Table 3.

### Caregivers' beliefs and practices of neonatal jaundice

A majority 84.7%, believe that jaundice is not a curse from the gods while 83.2% engaged in good practices by not putting their jaundiced babies in a dark room for at least 7 days. Also, 89.6% believe the yellowish color of their babies does not signify that their babies are growing well. Caregivers' beliefs and practices are shown in Table 4.

### Factors associated with knowledge of neonatal jaundice among caregivers

A multivariate ordered logistic regression analysis revealed that caregivers' who had prior awareness and education on neonatal jaundice were three times likely to exhibit a good knowledge on neonatal jaundice compared to caregivers without prior awareness and education on neonatal jaundice at a p = 0.001 [AOR = 3.02, (95%CI: 1.59–5.74), p = 0.001]. The factors associated with caregivers' knowledge is shown in Table 5.

### Factors associated with caregivers' attitude towards neonatal jaundice

In a multivariate logistic analysis, the odds of a having a good attitude regarding neonatal jaundice was two times more likely to occur among public service employers compared to those who were self/privately employed [AOR = 2.08, (95%CI: 1.03–4.21), p = 0.042]. Furthermore, caregivers who had one child were 58% less likely to demonstrate positive attitudes regarding

**Table 1. Socio-demographic characteristics of caregivers.**

| Variable | Frequency (f) | Percent (%) |
|---|---|---|
| **Age (years)** | | |
| 18–25 | 45 | 22.3 |
| 26–35 | 115 | 56.9 |
| 36–45 | 34 | 16.8 |
| >45 | 8 | 4.0 |
| **Sex** | | |
| Female | 187 | 92.6 |
| Male | 15 | 7.4 |
| **Religion** | | |
| Christian | 173 | 85.6 |
| Muslim | 22 | 10.9 |
| Traditional | 7 | 3.5 |
| **Marital status** | | |
| Single | 53 | 26.2 |
| Married | 131 | 64.9 |
| Divorced | 13 | 6.4 |
| Widowed | 5 | 2.5 |
| **Educational level** | | |
| None | 5 | 2.5 |
| Basic | 37 | 18.3 |
| Secondary | 69 | 34.2 |
| Tertiary | 91 | 45.0 |
| **Employment status** | | |
| Self/Private | 72 | 35.7 |
| Public Servant | 56 | 27.7 |
| Unemployed/Student | 14 | 6.9 |
| Unknown | 60 | 29.7 |
| **Ethnicity** | | |
| Ewe | 134 | 66.3 |
| Akan | 41 | 20.3 |
| Ga | 14 | 6.9 |
| Others | 13 | 6.5 |
| **Number of previous children** | | |
| None | 36 | 17.8 |
| One | 60 | 29.7 |
| Two | 63 | 31.2 |
| More than two | 43 | 21.3 |
| **Number of ANC visits** | | |
| <4 visits | 57 | 28.2 |
| 4+ visits | 145 | 71.8 |
| **Place of residence** | | |
| Urban | 115 | 56.9 |
| Rural | 72 | 35.7 |
| Estate | 15 | 7.4 |
| **Education received on neonatal jaundice** | | |
| No | 116 | 58 |
| Yes | 84 | 42 |

(*Continued*)

**Table 1.** (Continued)

| Variable | Frequency (f) | Percent (%) |
|---|---|---|
| **Child with a diagnosis of neonatal jaundice** | | |
| No | 173 | 86.5 |
| Yes | 27 | 13.5 |

neonatal jaundice [AOR = 0.42, (95%CI: 0.18–0.99), p = 0.049]. Factors associated with caregivers' attitude is shown in Table 6.

## Factors associated with caregivers' beliefs and practices of neonatal jaundice

Multivariate logistic regression analysis revealed that rural resident caregivers were 48% less likely to have good practices compared to urban residents [COR = 0.52 (95%CI: 0.29–0.93), p = 0.026]. Factors associated with caregivers' beliefs and practices are presented in Table 7.

## Discussion

The current study was undertaken to investigate knowledge, attitude and practice among caregivers' regarding neonatal jaundice. As a matter of importance, the reduction in neonatal mortality through the achievement of the health component of Sustainable Development Goals (SDGs) is imperative and timely, and one means of achieving this is through the creation of

## Summary of caregivers Knowledge, Attitude and Practice of Neonatal Jaundice

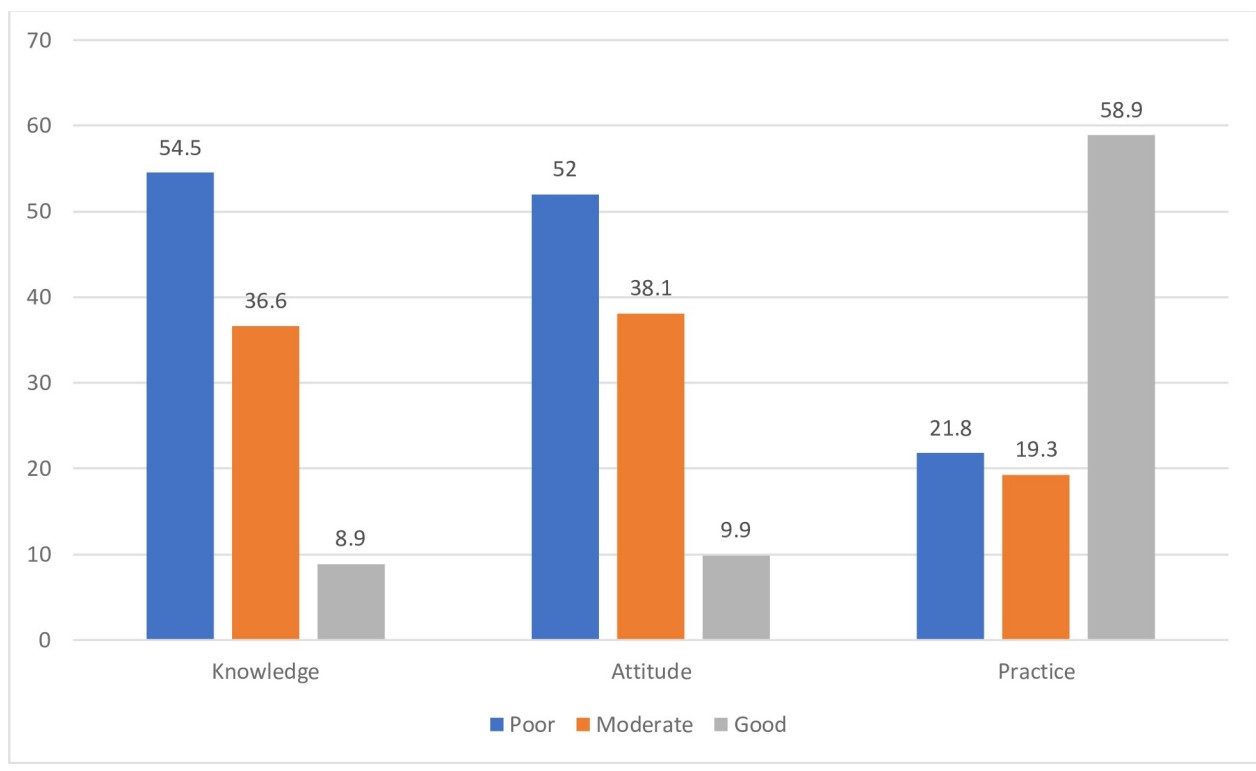

**Fig 1. Knowledge, attitude, and practice of caregivers on neonatal jaundice.**

**Table 2. Caregivers' knowledge about neonatal jaundice (n = 202).**

| Variable | Correct responses | |
|---|---|---|
| | Frequency (f) | Percent (%) |
| **Definition neonatal jaundice** | | |
| Jaundice is the yellowish discoloration of the skin and eyes | 181 | 89.6 |
| **Causes of neonatal jaundice** | | |
| Disparity between blood groups can cause jaundice | 70 | 34.7 |
| Prematurity of the baby can cause jaundice | 107 | 53.0 |
| Infection is a cause of jaundice | 103 | 51.0 |
| Feeding the baby with breastmilk can cause jaundice | 39 | 19.3 |
| Jaundice can be caused by giving your baby cold water | 94 | 46.5 |
| Delay passage of meconium can cause jaundice | 59 | 29.2 |
| **Complications of neonatal jaundice** | | |
| Jaundice can bring about brain damage in the baby | 83 | 41.1 |
| Jaundice can render a child physically handicapped | 92 | 45.5 |
| A baby with jaundice can develop convulsions | 112 | 55.5 |
| A baby diagnosed with jaundice can die | 157 | 77.7 |
| **Danger signs of neonatal jaundice** | | |
| A jaundiced baby feeds very poorly | 131 | 64.9 |
| Arching of the back is a danger sign of jaundice | 64 | 31.7 |
| Convulsion is a danger sign in a baby with jaundice | 101 | 50.0 |
| Refusal to eat is also a danger sign in a baby with jaundice | 99 | 49.0 |
| High pitch cry is a danger sign of jaundice | 65 | 32.2 |
| Down turning of the eye is a danger sign of jaundice in a baby | 80 | 39.6 |
| **Sites for checking neonatal jaundice** | | |
| The skin and eyes are sites for checking jaundice | 185 | 91.6 |
| The palms are also sites used to check jaundice | 168 | 83.2 |
| The urine of the baby is used to check for jaundice in a baby | 20 | 9.9 |
| Feaces of the child can be used to determine if the baby has jaundice or not | 38 | 18.8 |
| **Treatment of neonatal jaundice** | | |
| Phototherapy is one method used to treat neonatal jaundice | 92 | 45.5 |
| Exchange blood transfusion is also a method of treating neonatal jaundice | 78 | 38.6 |
| Going to church with frequent fasting and prayers are ways of treating neonatal jaundice | 111 | 55.0 |
| Traditional methods are also used to treat neonatal jaundice | 66 | 32.7 |
| **Prevention of neonatal jaundice** | | |
| **Neonatal Jaundice is a common problem on newborns so there is no need to prevent it** | 32 | 15.8 |
| Neonatal Jaundice can be prevented | 171 | 84.7 |
| Early initiation of breast milk can prevent neonatal jaundice | 125 | 61.9 |

Multiple response system was allowed

awareness, improving the knowledge, attitude and practices among caregivers. Improvement in maternal knowledge and early care-seeking behavior serve as fundamental components of effective management of neonatal jaundice and an implication for reducing neonatal mortality [11].

The current study reported that most caregivers have a poor knowledge regarding neonatal jaundice. Only 8.9% of the caregivers had good knowledge about neonatal jaundice. Comparing our findings to other results in some LMICs, it was realized that our 8.9% good knowledge

**Table 3. Attitude of caregivers towards neonatal jaundice.**

| Variable | Correct responses | |
|---|---|---|
| | Frequency (f) | Percent (%) |
| If I attend ANC frequently, I will receive education on the prevention and recognition of jaundice to help me prevent the condition | 164 | 81.2 |
| Jaundice in early life can be treated in the hospital | 174 | 86.1 |
| Herbal medications are used to treat jaundice | 44 | 21.8 |
| Breastfeeding is a means of treating my baby's jaundice | 32 | 15.8 |
| Expose of baby to early morning sunlight is a way of treating jaundice. | 61 | 30.2 |
| Jaundice in early life is caused by evil spirits | 144 | 71.3 |
| Poor personal and environmental hygiene causes jaundice | 75 | 37.1 |
| Exclusive breastfeeding prevents jaundice | 119 | 58.9 |
| Going to church frequently and engage in prayer and fasting are means of preventing jaundice | 120 | 59.4 |

Multiple response system was allowed

score was lower [15, 24, 27, 34]. However, despite an overall poor knowledge regarding neonatal jaundice in the current study, the majority of the caregivers demonstrated good knowledge regarding the definition and sites for checking for jaundice. These results largely corroborate similar findings in Ghana and other LMICs in which the majority of the caregivers correctly defined and stated the sites to identify neonatal jaundice [20, 25, 27, 35, 36]. The implication of good knowledge in the identification of jaundice is that caregivers will detect jaundice immediately after birth and take appropriate steps towards seeking prompt treatment in the hospital. Regarding the cause of neonatal jaundice, most caregivers in our study identified correctly prematurity and infection as the main causes which reflects a good knowledge. Contrary to this, several studies [8, 24, 25, 27] have reported significant knowledge gaps where 73%, 63%, 57.1% and 73.1% of mothers respectively indicated they did not know the cause of neonatal jaundice. The study of Amegan-Aho et al [25] in Ghana further reported that about 24.1% and 3.8% of mothers wrongly identified "consumption of too much oil" and mosquito bites as causes of neonatal jaundice despite more than 90% of the mother receiving ANC services. Mothers'

**Table 4. Traditional beliefs and practice of caregivers towards neonatal jaundice.**

| Variable | Correct response | |
|---|---|---|
| | Frequency (f) | Percent (%) |
| I believe jaundice is a curse from the gods | 171 | 84.7 |
| I will drop breast milk on my baby's eyes as a means of managing jaundice | 140 | 69.3 |
| I will not feed my baby with first breast milk as means to prevents jaundice | 158 | 78.2 |
| I drop seawater on my baby's eyes to help cure jaundice | 163 | 80.7 |
| I keep my baby away from light to help prevent jaundice | 143 | 70.8 |
| I cut the areas between my baby's eyebrow to help prevent jaundice | 147 | 72.8 |
| I put my jaundiced baby in the darkroom for at least 7 days | 168 | 83.2 |
| I believe jaundice makes a baby's skin look more beautiful | 176 | 87.1 |
| I believe the yellowish discoloration of the baby's skin is a good sign that the baby is growing healthy and beautiful | 181 | 89.6 |

Multiple response system was allowed

**Table 5. Logistic regression results showing factors associated with knowledge of neonatal jaundice among caregivers.**

| Variable | COR | 95%CI | p-value | AOR | 95%CI | p-value |
|---|---|---|---|---|---|---|
| **Age (years)** | | | | | | |
| 18–25 | | | | | | |
| 26–35 | 1.04 | 0.52–2.06 | 0.907 | | | |
| 36–45 | 1.10 | 0.45–2.68 | 0.835 | | | |
| >45 | 1.04 | 0.25–4.30 | 0.955 | | | |
| **Sex** | | | | | | |
| Female | | | | | | |
| Male | 3.29 | 1.21–8.91 | **0.019** | 2.64 | 0.90–7.66 | 0.074 |
| **Religion** | | | | | | |
| Christian | | | | | | |
| Muslim | 0.37 | 0.14–0.98 | **0.046** | 0.45 | 0.56–1.31 | 0.143 |
| Traditional | 0.39 | 0.07–2.03 | 0.267 | 0.28 | 0.05–1.68 | 0.164 |
| **Marital status** | | | | | | |
| Single | | | | | | |
| Married | 098 | 0.53–1.85 | 0.966 | | | |
| Divorced | 0.31 | 0.08–1.24 | 0.097 | | | |
| Widowed | 0.26 | 0.03–2.45 | 0.239 | | | |
| **Educational level** | | | | | | |
| None | | | | | | |
| Basic | 0.27 | 0.05–1.54 | 0.139 | | | |
| Secondary | 0.50 | 0.09–2.62 | 0.411 | | | |
| Tertiary | 1.40 | 0.27–7.18 | 0.688 | | | |
| **Employment status** | | | | | | |
| Self/Private | | | | | | |
| Public Servant | 2.46 | 1.23–4.92 | 0.011 | 1.83 | 0.88–4.07 | 0.102 |
| Unemployed/Student | 4.82 | 1.51–15.32 | 0.008 | 3.19 | 0.92–11.02 | 0.067 |
| **Ethnicity** | | | | | | |
| Ewe | | | | | | |
| Akan | 1.65 | 0.85–3.21 | 0.137 | | | |
| Ga | 1.20 | 0.42–3.45 | 0.729 | | | |
| Others | 0.37 | 0.10–1.37 | 0.136 | | | |
| **Number of previous children** | | | | | | |
| None | | | | | | |
| One | 0.40 | 0.17–0.89 | 0.025 | 0.56 | 0.23–1.35 | 0.197 |
| Two | 0.54 | 0.24–1.21 | 0.134 | 0.87 | 0.35–2.09 | 0.751 |
| More than two | 0.45 | 0.19–1.07 | 0.070 | 0.96 | 0.34–2.66 | 0.935 |
| **ANC visits** | | | | | | |
| <4 visits | | | | | | |
| 4+ visits | 0.60 | 0.33–1.09 | 0.092 | | | |
| **Residence** | | | | | | |
| Urban | | | | | | |
| Rural | 0.47 | 0.26–0.85 | **0.012** | 0.79 | 0.40–1.54 | 0.486 |
| Estate | 1.04 | 0.39–2.77 | 0.935 | 0.62 | 0.22–1.83 | 0.396 |
| **Previous neonatal jaundice education** | | | | | | |
| No | | | | | | |
| Yes | 2.93 | 1.67–5.13 | **<0.001** | 3.02 | 1.59–5.74 | **0.001** |
| **Neonatal jaundice** | | | | | | |

(*Continued*)

**Table 5.** (Continued)

| Variable | COR | 95%CI | p-value | AOR | 95%CI | p-value |
|---|---|---|---|---|---|---|
| No | | | | | | |
| Yes | 1.61 | 0.76–0.55 | 0.211 | | | |

Ordered logistic regression significant, p < 0.05

Odds ratios (OR) were all adjusted for possible confounding covariates

inability to identify the right causes of jaundice could be attributed to the low level of tertiary education as observed in their study. Nonetheless, the majority (76.0%) of mothers in their study demonstrated overall good awareness of neonatal jaundice than the current study. Importantly, a good knowledge in identifying the cause of jaundice is an indication that mothers will likely put in measures to prevent it and seek prompt treatment. Knowledge on the treatment of jaundice was low in our study. Despite the high literacy level of caregivers in this study, most of the caregivers reported going to church and engaging in fasting and prayers as means of treating neonatal jaundice. However, caregivers also correctly identified phototherapy and exchange blood transfusion as the main treatment modalities of jaundice. Though in the minority, the identification of phototherapy and exchange blood transfusion signified good knowledge, and this corroborates several other studies in LMICs [21, 22, 24, 37]. Having poor knowledge of identifying the cause of neonatal jaundice is a recipe for poor decision-making on the choice of treatment and a risk of not identifying avoidable factors that may lead to jaundice, and this was largely demonstrated in our study. Health education should emphasize phototherapy as the main treatment plan and the need to seek early care at the hospital when neonatal jaundice is detected. Furthermore, on the aspect of caregivers' knowledge regarding the complications of jaundice, the majority (77.7%) mentioned the death of the baby. The current figure is higher than the 57.8%, 55.1% and 71% reported in Nigeria and Malaysia respectively [26, 28, 38].

In a multivariate analysis, caregivers with prior awareness and education on neonatal jaundice were three times more likely to have good knowledge of jaundice than those who have never received education on jaundice. The study of Huq et al corroborates the current finding [28]. Though our study did not find an association between knowledge and caregivers' educational status, however, the majority had secondary and tertiary education which suggested that their knowledge on neonatal jaundice will be high. Shockingly, most of the caregivers demonstrated an overall poor knowledge regarding neonatal jaundice. This may be due to inadequate information given regarding jaundice which may be a recipe for the poor knowledge observed or it may as well be as a result of poor health-seeking behavior on the part of the caregivers. The poor knowledge could also be due to low health literacy even in the midst of a high level of education. Collaborated effort is needed from healthcare professionals to impart knowledge on neonatal jaundice among caregivers during ANC visits using recommended guidelines as provided by the Ministry of Health GHS.

Findings from the current study revealed that most of the caregivers demonstrated a poor attitude towards neonatal jaundice. Despite the poor attitude, some of the caregivers correctly responded that they will not expose their babies to sunlight and will not use herbal medications to treat jaundice. Though these were positive responses, the majority will indulge in negative practices which signifies poor attitude. The practice of exposing jaundiced babies to sunlight is however common in Africa and other developing countries which explains the reason for its practice in our study [19, 20, 26, 28, 38, 39]. The majority of caregivers in our study will visit ANC frequently to prevent jaundice and most will send a jaundice baby to the hospital for

**Table 6. Logistic regression results showing factors associated with caregivers' attitude towards neonatal jaundice.**

| Variable | COR | 95%CI | p-value | AOR | 95%CI | p-value |
|---|---|---|---|---|---|---|
| **Age (years)** | | | | | | |
| 18–25 | | | | | | |
| 26–35 | 1.37 | 0.70–2.70 | 0.356 | | | |
| 36–45 | 1.51 | 0.63–3.62 | 0.553 | | | |
| >45 | 1.17 | 0.28–4.79 | 0.830 | | | |
| **Sex** | | | | | | |
| Female | | | | | | |
| Male | 0.70 | 0.24–1.98 | 0.497 | | | |
| **Religion** | | | | | | |
| Christian | | | | | | |
| Muslim | 0.58 | 0.23–1.44 | 0.241 | | | |
| Traditional | 0.16 | 0.02–1.33 | 0.090 | | | |
| **Marital status** | | | | | | |
| Single | | | | | | |
| Married | 0.85 | 0.46–1.56 | 0.615 | | | |
| Divorced | 0.38 | 0.11–1.33 | 0.130 | | | |
| Widowed | 0.54 | 0.09–2.67 | 0.504 | | | |
| **Educational level** | | | | | | |
| None | | | | | | |
| Basic | 1.04 | 0.16–6.60 | 0.964 | | | |
| Secondary | 1.65 | 0.27–9.97 | 0.581 | | | |
| Tertiary | 1.77 | 0.30–10.38 | 0.528 | | | |
| **Employment status** | | | | | | |
| Self/Private | | | | | | |
| Public Servant | 2.17 | 1.09–4.31 | **0.027** | 2.08 | 1.03–4.21 | **0.042** |
| Unemployed/Student | 2.98 | 1.05–8.42 | **0.039** | 2.37 | 0.76–7.35 | 0.535 |
| **Ethnicity** | | | | | | |
| Ewe | | | | | | |
| Akan | 0.71 | 0.36–1.41 | 0.329 | | | |
| Ga | 0.68 | 0.23–2.02 | 0.491 | | | |
| Others | 0.38 | 0.11–1.28 | 0.119 | | | |
| **Number of children** | | | | | | |
| None | | | | | | |
| One | 0.34 | 0.15–0.76 | **0.009** | 0.42 | 0.18–0.99 | **0.049** |
| Two | 0.60 | 0.28–1.32 | 0.206 | 0.77 | 0. 33–1.77 | 0.535 |
| More than two | 0.51 | 0.22–1.18 | 0.116 | 0.67 | 0.27–1.64 | 0.383 |
| **ANC visits** | | | | | | |
| <4 visits | | | | | | |
| 4+ visits | 0.90 | 0.50–1.62 | 0.728 | | | |
| **Residence** | | | | | | |
| Urban | | | | | | |
| Rural | 0.90 | 0.51–1.59 | 0.712 | | | |
| Estate | 1.13 | 0.38–3.31 | 0.829 | | | |
| **neonatal jaundice education** | | | | | | |
| No | 1.28 | 0.74–2.18 | 0.386 | | | |
| Yes | | | | | | |
| **Neonatal jaundice** | | | | | | |

*(Continued)*

**Table 6.** (Continued)

| Variable | COR | 95%CI | p-value | AOR | 95%CI | p-value |
|---|---|---|---|---|---|---|
| No | | | | | | |
| Yes | 0.82 | 0.38–1.82 | 0.635 | | | |

Ordered logistic regression significant, p < 0.05

Odds ratios (OR) were all adjusted for possible confounding covariates

treatment which is an indication of a good attitude. Other studies [20, 34, 40, 41] corroborates this finding in other resource-limited settings. A positive attitude implies that it will lead to early detection and correct diagnosis as well as prompt treatment with phototherapy or exchange blood transfusion with the ultimate aim of reducing neonatal mortality. The negative attitudes of exposing babies to the sunlight and the use of herbal preparation need to be discouraged through appropriate education during ANC as these are inimical to the survival of the babies and may lead to delays in early care-seeking at the hospital which may result in complications and even death.

Beliefs and practices regarding neonatal jaundice among the caregivers were good in this study. On the aspect of the belief questions, the majority of the caregivers answered positively. However, despite the positive responses regarding their practices, caregivers also responded negatively. For instance, about 30.3% of caregivers indicated that they will put breast milk in the baby's eyes as a means of managing jaundice, 29.2% will keep babies away from light to help prevent jaundice while 27.2% will cut the areas between the baby's eyebrow to help prevent jaundice.

Multivariate logistic regression analysis revealed that those who resided in rural areas were less likely to have good practices of neonatal jaundice compared to urban residents. This could probably be possible due to the availability of adequate health facilities and health care professionals in the urban areas who offer frequent education to caregivers at health facilities thereby improving their attitudes. Also, the majority of the caregivers resided in urban areas.

## Limitation of the study

First of all, the study was conducted in a single health facility making the findings not generalizable. Secondly, the study focused only on real parents of the babies and did not include significant others such as grandmothers, sisters or aunts etc. Also, the study excluded adolescent mothers/fathers who could have provided more revealing information about neonatal jaundice. Despite the above limitations, the study provided an insight into the caregivers' perspectives regarding jaundice and adds to the body of knowledge that will help reform newborn care practices in Ghana.

## Conclusion

More than 50% of the caregivers demonstrated overall poor knowledge and attitude regarding neonatal jaundice while about 58.9% had good practices. That notwithstanding, some caregivers demonstrated good knowledge regarding neonatal jaundice including but not limited to defining jaundice as the yellowish coloration of eyes and skin, identifying prematurity as a cause of jaundice, and indicating that jaundice in the newborn can be prevented. Contrary to the good knowledge, some indicated that going to church and engaging in fasting and prayers were means of treating jaundice which depict poor knowledge. Despite the overall good belief and practice score seen this study, caregivers will put breast milk in the baby's eyes to treat

**Table 7. Logistic regression results showing factors associated with beliefs and practices of caregivers regarding neonatal jaundice.**

| Variable | COR | 95%CI | p-value |
|---|---|---|---|
| **Age (years)** | | | |
| 18–25 | | | |
| 26–35 | 1.40 | 0.70–2.79 | 0.341 |
| 36–45 | 1.19 | 0.51–2.80 | 0.692 |
| >45 | 1.34 | 0.29–6.20 | 0.705 |
| **Sex** | | | |
| Female | | | |
| Male | 2.19 | 0.69–6.96 | 0.184 |
| **Religion** | | | |
| Christian | | | |
| Muslim | 1.30 | 0.54–3.16 | 0.561 |
| Traditional | 0.30 | 0.06–1.43 | 0.133 |
| **Marital status** | | | |
| Single | | | |
| Married | 1.39 | 0.75–2.59 | 0.298 |
| Divorced | 0.78 | 0.23–2.59 | 0.682 |
| Widowed | 0.90 | 0.13–6.06 | 0.917 |
| **Educational level** | | | |
| None | | | |
| Basic | 0.29 | 0.03–2.81 | 0.283 |
| Secondary | 0.31 | 0.03–2.99 | 0.314 |
| Tertiary | 0.58 | 0.06–5.50 | 0.636 |
| **Employment status** | | | |
| Self/Private | | | |
| Public Servant | 1.56 | 0.79–3.11 | 0.201 |
| Unemployed/Student | 1.57 | 0.49–5.03 | 0.445 |
| **Ethnicity** | | | |
| Ewe | | | |
| Akan | 1.08 | 0.53–2.20 | 0.836 |
| Ga | 1.04 | 0.33–3.25 | 0.956 |
| Others | 0.62 | 0.24–1.60 | 0.325 |
| **Number of children** | | | |
| None | | | |
| One | 0.62 | 0.27–1.40 | 0.251 |
| Two | 1.04 | 0.45–2.39 | 0.926 |
| More than two | 0.76 | 0.54–1.84 | 0.536 |
| **ANC visits** | | | |
| <4 visits | | | |
| 4+ visits | 0.85 | 0.46–1.58 | 0.608 |
| **Residence** | | | |
| Urban | | | |
| Rural | 0.52 | 0.29–0.93 | **0.026** |
| Estate | 0.45 | 0.15–1.34 | 0.151 |
| **neonatal jaundice education** | | | |
| No | | | |
| Yes | 1.30 | 0.74–2.26 | 0.356 |

(*Continued*)

**Table 7.** (Continued)

| Variable | COR | 95%CI | p-value |
|---|---|---|---|
| **Neonatal jaundice** | | | |
| No | | | |
| Yes | 1.42 | 0.61–3.28 | 0.414 |

jaundice while others will cut the areas between the baby's eyebrow to help prevent jaundice and a few of them indicating correctly that traditional methods are not used to treat neonatal jaundice.

Caregivers with prior awareness and education of neonatal jaundice were more likely to demonstrate good knowledge of neonatal jaundice. Also, caregivers employed in the public sector were more likely to have a good attitude towards neonatal jaundice while those who resided in rural were less likely to have good practices.

In the light of the nation's dire need to reduce neonatal mortality rates which is in line with the health aspects of the United Nations SDGs, creating awareness, improving knowledge and attitudes and dispelling misconceptions about neonatal jaundice is timely. Therefore, concerted effort is required from frontline healthcare professionals to collaboratively intensify the education on jaundice in the areas of early identification of cases, causes, treatment, danger signs and prevention.

## Supporting information

**S1 Dataset.**
(XLSX)

## Acknowledgments

The authors express their profound gratitude to the caregivers for allowing time to take part in this study. Further gratitude goes to the management of the Ho Teaching Hospital for permitting us to conduct the study in the hospital.

## Author Contributions

**Conceptualization:** Solomon Mohammed Salia.

**Data curation:** Solomon Mohammed Salia, Agani Afaya, Abubakari Wuni, Martin Amogre Ayanore, Emmanuel Salia, Doreen Dzidzor Kporvi, Peter Adatara, Vida Nyagre Yakong, Sean Augustine Eduah-Quansah, Shine Seyram Quarshie, Eric Kwame Dey, Dominic Amoah Akolga, Robert Kaba Alhassan.

**Formal analysis:** Solomon Mohammed Salia, Agani Afaya, Martin Amogre Ayanore, Robert Kaba Alhassan.

**Funding acquisition:** Solomon Mohammed Salia, Agani Afaya, Abubakari Wuni, Martin Amogre Ayanore, Doreen Dzidzor Kporvi, Peter Adatara, Sean Augustine Eduah-Quansah, Shine Seyram Quarshie, Eric Kwame Dey, Dominic Amoah Akolga, Robert Kaba Alhassan.

**Investigation:** Solomon Mohammed Salia, Agani Afaya, Abubakari Wuni, Martin Amogre Ayanore, Doreen Dzidzor Kporvi, Peter Adatara, Sean Augustine Eduah-Quansah, Shine Seyram Quarshie, Eric Kwame Dey, Dominic Amoah Akolga, Robert Kaba Alhassan.

**Methodology:** Solomon Mohammed Salia, Agani Afaya, Abubakari Wuni, Martin Amogre Ayanore, Peter Adatara, Robert Kaba Alhassan.

**Project administration:** Solomon Mohammed Salia, Agani Afaya, Abubakari Wuni, Martin Amogre Ayanore, Emmanuel Salia, Doreen Dzidzor Kporvi, Peter Adatara, Sean Augustine Eduah-Quansah, Shine Seyram Quarshie, Eric Kwame Dey, Dominic Amoah Akolga, Robert Kaba Alhassan.

**Resources:** Solomon Mohammed Salia.

**Software:** Solomon Mohammed Salia, Martin Amogre Ayanore, Peter Adatara, Robert Kaba Alhassan.

**Supervision:** Solomon Mohammed Salia.

**Validation:** Solomon Mohammed Salia, Martin Amogre Ayanore, Peter Adatara, Vida Nyagre Yakong, Robert Kaba Alhassan.

**Visualization:** Solomon Mohammed Salia, Agani Afaya, Abubakari Wuni, Martin Amogre Ayanore, Emmanuel Salia, Doreen Dzidzor Kporvi, Peter Adatara, Vida Nyagre Yakong, Sean Augustine Eduah-Quansah, Shine Seyram Quarshie, Eric Kwame Dey, Dominic Amoah Akolga, Robert Kaba Alhassan.

**Writing – original draft:** Solomon Mohammed Salia.

**Writing – review & editing:** Solomon Mohammed Salia, Agani Afaya, Abubakari Wuni, Martin Amogre Ayanore, Emmanuel Salia, Doreen Dzidzor Kporvi, Peter Adatara, Vida Nyagre Yakong, Sean Augustine Eduah-Quansah, Shine Seyram Quarshie, Eric Kwame Dey, Dominic Amoah Akolga, Robert Kaba Alhassan.

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
