## [Decision Letter · Decision Letter 0]

27 Nov 2020

PONE-D-20-19725

Determinants of Neonatal Jaundice Knowledge, Attitudes and Practice Among Caregivers in a Tertiary Health Facility in Ghana: Implications for Neonatal Care policy Reforms and Reducing Neonatal Mortality in a Low Resource Setting.

PLOS ONE

Dear Dr. Salia,

Thank you for submitting your manuscript to PLOS ONE. After careful consideration, we feel that it has merit but does not fully meet PLOS ONE’s publication criteria as it currently stands. Therefore, we invite you to submit a revised version of the manuscript that addresses the points raised during the review process.

I would like to sincerely apologise for the delay you have incurred with your submission. It has been exceptionally difficult to secure reviewers to evaluate your study. We have now received two completed reviews; their comments are available below.

Please revise the manuscript to address all the reviewer's comments in a point-by-point response in order to ensure it is meeting the journal's publication criteria. Please note that the revised manuscript will need to undergo further review, we thus cannot at this point anticipate the outcome of the evaluation process.

We look forward to receiving your revised manuscript.

Kind regards,

Miquel Vall-llosera Camps

Senior Editor

PLOS ONE

Journal Requirements:

2. Please include a separate caption for each figure in your manuscript.

**Comments to the Author**

1. Is the manuscript technically sound, and do the data support the conclusions?

Reviewer #1: Partly

Reviewer #2: Yes

2. Has the statistical analysis been performed appropriately and rigorously? 

Reviewer #1: Yes

Reviewer #2: I Don't Know

3. Have the authors made all data underlying the findings in their manuscript fully available?

Reviewer #1: No

Reviewer #2: Yes

4. Is the manuscript presented in an intelligible fashion and written in standard English?

Reviewer #1: Yes

Reviewer #2: Yes

5. Review Comments to the Author

Reviewer #1: Determinants of Neonatal Jaundice Knowledge, Attitudes and Practice among Caregivers in Ghana

Introduction and Literature Review:

Neonatal Jaundice is an important topic in the West African sub-region, the significance of jaundice being emphasized by the complications that attend uncontrolled rise in unconjugated hyperbilirubinaemia . Jaundice may progress to acute bilirubin encephalopathy with hearing loss and cerebral palsy or death besides lingering kernicterus spectrum disorder. The author needs to elaborate this outcome of jaundice because it is ruinous to families that become unfortunate to experience the disorder.

Jaundice occurs in physiological or pathological forms. While the former is harmless the latter is deleterious. What is concerning is the pathological jaundice which may lead to neonatal mortality. Literature review however scanty it may be needs to cover the entire understanding of the subject matter as the KAP of parents may be influenced by this knowledge.

Study Population:

The study focused on the real fathers/mothers of the neonates excluding other significant patient relatives. In the study location of West Africa, a good number of caregivers are neither the real father/mother of the neonate. Very often the grandmother, aunt, sister or other significant relative play an important, sometimes dominant role in neonatal care. Thus excluding them from a study of this nature could mean a major information loss and interventions targeted at real mothers/fathers may fall short of success without them. In the author’s area of practice the grandmothers often overrule their daughters in child care practices irrespective of the daughter’s knowledge or educational background.

Inclusion/Exclusion Criteria:

The author excluded adolescent mothers/fathers from the study. In West Africa, teenage pregnancy and delivery is common. These mothers are inexperienced and therefore more likely to be ignorant of good child care practices. Studies of under-five mortality show that infants born to these teenagers succumb by far more than the experience mothers of older age group. This study would have been more revealing and perhaps more beneficial if adolescent mothers (12-18 years) were included in the study.

Study Design:

The study design was entirely quantitative but the issues were more dominantly qualitative. A study which set out to evaluate the KAP of respondents should have given them the opportunity to express themselves freely in order to optimize that information that the researcher may not have thought about. This is a major concern in this study. How can we be sure that the investigators captured the reflections of the respondents’ true inclination?

Data Collection procedure:

The sampling of study participants was by convenient method. How could this methodology have navigated around bias? The questionnaire was administered to two groups differently, to one group by self-administration while to the second group it was by interviewer administration. A number of the respondents had no formal education requiring interpretation of the questionnaire. How exactly was this bottle neck sorted out? Was translation into local languages carried out? Please specify the details.

Assessment of Knowledge of mothers/caregivers on NJ:

It is eminently observable that death as an outcome of untreated NJ was not evaluated in this study, neither as a complication nor as danger sign. Could knowledge of this adverse outcome influence respondents’ attitude and change their health seeking behavior? If this was evaluated, it would be important to analyze the data to enrich the present study.

Treatment of NJ:

Several treatment modalities were indicated by the authors in evaluating the KAP of the respondents. All modes of treatment were investigated individually. Were there respondents who practiced a combination of more than one mode of treatment of NJ in the study? Fifty-five respondents volunteered that they sought help by visiting “church” or by “fasting and praying”. How many of these respondents also visited the health facility for treatment of the NJ? It is possible to criminalize a respondent’s practice without fully exploring all the actions taken towards treatment of their neonates’ NJ. In West Africa usually a combination of approaches are resorted to in order to overcome health challenges.

Attitude of Caregivers towards NJ:

An overwhelming number of the respondents demonstrated the attitude of attending ANC and a large number of respondents showed awareness that NJ early in life can be treated in the hospital. However, the investigators did not evaluate how many of the respondents actually practiced taking their infants to the hospital for treatment of NJ. In West Africa most health care seekers would apply traditional methods of treatment of ailments and would also visit the hospital for treatment. Was it a deliberate action not to include this point or an omission?

Confounding variables:

KAP may be influenced by other considerations that may complicate the ones studied herein. Such other factors include health infrastructure and availability of equipment, economic realities, and distance to the health facility, availability of transportation and insecurity. How were these confounding variable eliminated so we can trust the validity of this study findings?

Reviewer #2: Reviewer:

Plos One

Title: Determinants of Neonatal Jaundice Knowledge, Attitudes and Practice Among Caregivers in a Tertiary Health Facility in Ghana: Implications for Neonatal Care policy Reforms and Reducing Neonatal Mortality in a Low Resource Setting.

Date: 26/11/2020

General comment:

Thank you for allowing me to review this article by Salia, et al. The manuscript is about neonatal jaundice a common condition affecting newborns which can lead to disability and sometimes mortality. Generally, the manuscript reads well and is of relevance to those with an interest in the public health aspect of newborn care. However there are several abbreviations such as Neonatal Jaundice, neonatal mortality and postnatal which are simple words or phrases, that can be obtrusive to reading and a few other items that need to be corrected prior to publication. I have made some suggestions below:

Major corrections

• Page 1: Title: Suggest revision of title to: Knowledge, Attitudes and Practices Regarding Neonatal Jaundice Among Caregivers in a Tertiary Health Facility in Ghana.

• This was what was done. The study is a small study and finding are not so different from what is already known or so novel to warrant a reform in policy.

• Page 6 Sample size and sampling determination (change determination to methods) - it is not clear what sampling method was actually applied. While is says in this section that systematic sampling was done, on Page 8 line 10 it says ...caregivers were conveniently selected – this is in contrast with the assertion on page 6 that systematic sampling was used. Please clarify.

• Page 6 Line 12 and 13: was selection of the first skip interval done by balloting or other random sampling method? How were the patients arranged ina row? Were they arranged according to time of arrival as one would imagine would be the same as the attendance records or in another way? This needs to be specified.

• Page 2 abstract line 12 under methods: It does not say where the study was done and what data was collected.

Minor essential corrections

• Abstract

• Page 2 line 23: suggest rephrase to “good attitude about neonatal jaundice” rather than “...attitude in jaundice”.

• Page 2 line 29: suggest rephrase to “,,, will help reduce disability and deaths from neonatal jaundice.”

Introduction

• Page 3 line 14: it will be useful to also capture global targets (SDG)

• Page 3 lines 9-11. Although neonatal jaundice affects a number of babies, many of them recover. A few may die from kernicterus or bilirubin encephalopathy however underlying causes such as neonatal sepsis are more often the cause of neonatal death in babies with jaundice. Nonetheless neonatal jaundice is a major cause of disability i.e. cerebral palsy and emphasis of this role is lacking. Thus need to rephrase to capture this.

• Page 4 line 9: should read “...as a means of treatment”

• Page 4 line 10 change “generations back” to “previous generations”

• Page 4 line18 suggest rephrasing to read...serve as useful information

Methods

• Page 5 ethical approval: were the benefits and risks also explained to them?

• Study area: Information on the delivery rate, presence of a newborn unit, its capacity, admission and neonatal mortality rates will be useful.

• Page 5 line 17: ...the health facility serves as a major facility for neighbouring Togo residents – needs to be put in the right perspective as unintentional.

• Page 5 line 17 bed capacity” remove approximately- be specific

• Page 6 study design – reads better without the abbreviations. i.e. if they are written in full -. KAP and NJ

• Page 6 line 20 - 21 Inclusion criteria- need to include the recruitment period

• Page 7 line 1: rephrase to “...not the parents of the baby” rather than “...not the real father or mother”

• Page 7 line 7 - no mention of northern tribes – they form a significant proportion of the Ghanaian population, were they few?

• Page 7 line 12: Adequacy of antenatal visits is now pegged at 8 or more visits but 4 or more can still be used as information on this is more readily available for comparison. Need to mention

• Page 7 line 15 -17. Add - these are captured in more details in the results section .

• Page 8 line 17: “...further explanation” seems a more appropriate to understanding

• Page 9 line 4 comment of significance of these coefficients.

Results

• Page 10. Table 1 title: choose caregivers or participants

• Page 13: frequent ANC attendance alone will not prevent jaundice unless it provides education on prevention and recognition. Sentence should reflect this.

• Page 13 Lines 4 and 5 only a few identified traditional practices as being wrong should be reflected in the discussion and conclusion

• Page 10. Table 1 an unusually large number of caregivers with tertiary education needs an explanation and additional information on catchment area for hospital or was there a selection before the sampling method was applied

• Page 13 Attitude questions seem like knowledge questions.

• Page 13 last but one sentence spelling of colour –color

• Page 17 last line & Table 7 suggests - factors associated rather than determinants Having one child was associated p-0.009 (with) ones practices, but not explained or discussed.

• Conclusion poor knowledge regarding traditional treatments not mentioned

• Discussion

• Page 20 line 22: Is it possible that the reason for the disparity between your finding and Amegan-Aho et al‘s is because there were several caregivers with tertiary education in this study?

• Page 21 line 6 suggest “...though in the minority...”

• Page 21 line 8 suggest “... poor knowledge of (omit “a”)”

• Page 21 line 14 suggest “... knowledge of... rather than “in”.

• Page 21 last line: what is meant by “positive” knowledge?

• Page 22 last but 3 line suggest removal of “will” to read who “offer”

Discretionary corrections

• Page 4 line 7 suggest revising “where” to read “because”.

• Page 8 line 11: questions asked in the questionnaire –suggest rephrase

• Page 20 Line 12: suggest revising “...where majority to “in which majority”

• Page 21 line 17 suggest removal of “pre” before suggested

6. PLOS authors have the option to publish the peer review history of their article (what does this mean?). If published, this will include your full peer review and any attached files.

Reviewer #1: No

Reviewer #2: No

---

## [Author Response · Author response to Decision Letter 0]

6 Jan 2021

Dear editor,

We appreciate your efforts in helping us to shape our manuscript to the required standard for publication in your reputable journal. We would like to thank the reviewer for the insightful comments on our manuscript. We are particularly grateful for the opportunity to resubmit the manuscript for another round of review. We find the comments very useful and have responded to them to the best of our knowledge. We acknowledge that the comments have no doubt helped improve the quality of our manuscript.

We have therefore provided further details by showing point-by-point feedback on how each of the comments by the reviewer were addressed. For easy identification of our responses, the reviewers’ comments have been repeated while the Authors’ responses appear in BOLD text in the main manuscript.

REVIEWER #1

Introduction and Literature Review:

Neonatal Jaundice is an important topic in the West African sub-region, the significance of jaundice being emphasized by the complications that attend uncontrolled rise in unconjugated hyperbilirubinaemia. Jaundice may progress to acute bilirubin encephalopathy with hearing loss and cerebral palsy or death besides lingering kernicterus spectrum disorder. The author needs to elaborate this outcome of jaundice because it is ruinous to families that become unfortunate to experience the disorder.

Jaundice occurs in physiological or pathological forms. While the former is harmless the latter is deleterious. What is concerning is the pathological jaundice which may lead to neonatal mortality. Literature review however scanty it may be needs to cover the entire understanding of the subject matter as the KAP of parents may be influenced by this knowledge.

Authors response

Page 3 line 76-91: The authors agreed that the above comment is relevant in the area of neonatal jaundice as it will reveal the deleterious effects of jaundice to the reader. Given this, the introduction section of the manuscript has been revised to address this comment.

Study Population:

The study focused on the real fathers/mothers of the neonates excluding other significant patient relatives. In the study location of West Africa, a good number of caregivers are neither the real father/mother of the neonate. Very often the grandmother, aunt, sister or other significant relative play an important, sometimes dominant role in neonatal care. Thus excluding them from a study of this nature could mean a major information loss and interventions targeted at real mothers/fathers may fall short of success without them. In the author’s area of practice the grandmothers often overrule their daughters in child care practices irrespective of the daughter’s knowledge or educational background.

Authors response

The authors appreciate the in-depth review carried out on the manuscript and the study subject by the reviewer in relation to the geographic area of the study. We acknowledged that the addition of these issues by the reviewer could by far improved the quality of the manuscript. However, these issues escaped us when we were designing the study and we have therefore included this comment as part of the limitations of the study. In subsequent related studies, we will take cognizance of these comments. 

Inclusion/Exclusion Criteria:

The author excluded adolescent mothers/fathers from the study. In West Africa, teenage pregnancy and delivery is common. These mothers are inexperienced and therefore more likely to be ignorant of good child care practices. Studies of under-five mortality show that infants born to these teenagers succumb by far more than the experience mothers of older age group. This study would have been more revealing and perhaps more beneficial if adolescent mothers (12-18 years) were included in the study.

Authors response

We greatly agree with the reviewer on this comment that the inclusion of these groups of mothers could have brought out information/findings that would have contributed to policy reforms in the area of newborn care in Ghana. Despite the omission, we believe the study has revealed some key findings that will add to the body of knowledge and worthy of influencing newborn care positively in Ghana. We have therefore revised the manuscript to include this comment as part its limitation on page 25.

Study Design:

The study design was entirely quantitative but the issues were more dominantly qualitative. A study which set out to evaluate the KAP of respondents should have given them the opportunity to express themselves freely in order to optimize that information that the researcher may not have thought about. This is a major concern in this study. How can we be sure that the investigators captured the reflections of the respondents’ true inclination?

Authors response

The authors reviewed a wide range of relevant literature regarding this subject area which were equally done as quantitative studies and had included a wide range of relevant questions from literature and questions arising from discussions with some mothers. We however agree that allowing the mothers to express themselves freely on such an important topic would have revealed more information. We therefore believe that the questions asked in our study are by far the reflections of the caregivers’ true inclination. Subsequently, a mixed-method could be considered in related studies.

Data Collection procedure:

The sampling of study participants was by convenient method. How could this methodology have navigated around bias? The questionnaire was administered to two groups differently, to one group by self-administration while to the second group it was by interviewer administration. A number of the respondents had no formal education requiring interpretation of the questionnaire. How exactly was this bottle neck sorted out? Was translation into local languages carried out? Please specify the details.

Authors response

The method of sampling in this study was a systematic random sampling as found in the abstract and on page 2 line 52 of the manuscript. The manuscript has been revised to remove the convenient method from the manuscript as this was purely a mistake. The use of the systematic random sampling method eliminated any possibility of bias as a sampling interval was obtained to be (2). Participants were arranged to sit in rows and the first respondent who formed the starting point was selected by balloting. This is found on page 7 line 180-184.

Respondents who were proficient in speaking and writing in the English language answered the questionnaires by themselves. For those who could neither read nor write in the English language were guided to complete the questionnaire. Four of the authors were native speakers of the “Ewe” language and three authors who could speak “Ewe”, “Twi” and other languages were used to orally translate the questionnaire into these local languages for the understanding of the caregivers. This has subsequently been included in the data collection procedure section of the manuscript on page 10 line 237-242.

Assessment of Knowledge of mothers/caregivers on NJ:

It is eminently observable that death as an outcome of untreated NJ was not evaluated in this study, neither as a complication nor as danger sign. Could knowledge of this adverse outcome influence respondents’ attitude and change their health seeking behavior? If this was evaluated, it would be important to analyze the data to enrich the present study.

Authors response

The authors find the comment very useful. Indeed, knowing that poor health seeking behaviour could lead to the death of a child could definitely influence caregivers attitude and their health-seeking behaviours. However, on page 15 (table 2: caregivers knowledge of neonatal jaundice) under the complication section, death has been captured where the majority (77.7%) of caregivers indicated that death could occur as a result of jaundice. We have further included this in the discussion section of the manuscript on page 24 line 422-424.

Treatment of NJ:

Several treatment modalities were indicated by the authors in evaluating the KAP of the respondents. All modes of treatment were investigated individually. Were there respondents who practiced a combination of more than one mode of treatment of NJ in the study? Fifty-five respondents volunteered that they sought help by visiting “church” or by “fasting and praying”. How many of these respondents also visited the health facility for treatment of the NJ? It is possible to criminalize a respondent’s practice without fully exploring all the actions taken towards treatment of their neonates’ NJ. In West Africa usually a combination of approaches are resorted to in order to overcome health challenges.

Authors response

We appreciate this comment. The authors allowed multiple responses from the caregivers. We therefore believed that the caregivers responded multiple times by choosing responses that applied to them.

Attitude of Caregivers towards NJ:

An overwhelming number of the respondents demonstrated the attitude of attending ANC and a large number of respondents showed awareness that NJ early in life can be treated in the hospital. However, the investigators did not evaluate how many of the respondents actually practiced taking their infants to the hospital for treatment of NJ. In West Africa most health care seekers would apply traditional methods of treatment of ailments and would also visit the hospital for treatment. Was it a deliberate action not to include this point or an omission?

Authors response

The authors agreed with the reviewer on this comment that most healthcare seekers would apply traditional methods of treatment of ailments and would also visit the hospital for treatment. However, it was purely an oversight on the part of the authors not to have included whether “caregivers actually took their infants to the hospital for treatment of neonatal jaundice” in our study. We appreciate that the addition of this to our study would have revealed more attitudes from the caregivers. 

Confounding variables:

KAP may be influenced by other considerations that may complicate the ones studied herein. Such other factors include health infrastructure and availability of equipment, economic realities, and distance to the health facility, availability of transportation and insecurity. How were these confounding variables eliminated so we can trust the validity of this study findings?

Authors response

This study investigated knowledge, attitude and practice of neonatal jaundice among caregivers. The authors agreed that newborn care practices may be influenced by the factors stated in the reviewer’s comment. However, our questionnaire was designed based on the main aim of the study which was to evaluate caregivers knowledge, attitude and practice of neonatal jaundice. The questionnaire went through a thorough validity process where it was peer-reviewed by all the authors, nursing and medical specialists in related field to the study topic as well as piloting and pretesting. This made it possible that all questions included in the study were relevant and specific in achieving the intended purpose of the study.

Also, the authors carried out multicollinearity diagnostics and found the Variance Inflation Factors (VIFs) were all below the 5-10 rule of thumb range, suggesting there is no collinearity among the independent variables to be fitted in the regression model! Indeed.

REVIEWER #2

MAJOR CORRECTIONS 

Reviewer’s comment

Page 1: Title: Suggest revision of title to: Knowledge, Attitudes and Practices Regarding Neonatal Jaundice Among Caregivers in a Tertiary Health Facility in Ghana.

Authors response

Page 1 line 2-3: We agree with the reviewer that the main outcome variables of this study were knowledge, attitude and practice and that the study size was small to influence policy. We have therefore revised the title of the manuscript to reflect the reviewer’s comment.

Reviewer’s comment

Page 6 Sample size and sampling determination (change determination to methods) - it is not clear what sampling method was actually applied. While it says in this section that systematic sampling was done, on Page 8 line 10 it says ...caregivers were conveniently selected – this is in contrast with the assertion on page 6 that systematic sampling was used. Please clarify. 

Authors response

Page 7 line 175-178: The manuscript was formatted according to the journal specifications, however, because of the comment, “sampling determination” has been revised to “sampling method”. The sampling method for this study was systematic random sampling. Therefore, the manuscript was revised to remove “caregivers were conveniently selected” to reflect the correct sampling method as stated under sample size and sampling method. 

Reviewer’s comment

Page 6 Line 12 and 13: was selection of the first skip interval done by balloting or other random sampling method? How were the patients arranged in a row? Were they arranged according to time of arrival as one would imagine would be the same as the attendance records or in another way? This needs to be specified.

Authors response

Page 7 line 170-184: The authors have revised this section of the manuscript to address these concerns. 

Reviewer’s comment

Page 2 abstract line 12 under methods: It does not say where the study was done and what data was collected.

Authors response

Page 2 line 51-53: The method section of the abstract was revised to include where the study was done and the data collected. 

MINOR ESSENTIAL CORRECTIONS

ABSTRACT

Reviewer’s comment

Page 2 line 23: suggest rephrase to “good attitude about neonatal jaundice” rather than “...attitude in jaundice”.

Authors response

Page 2 line 61-62: The authors revised the abstract to include the reviewer’s suggestion.

Reviewer’s comment

Page 2 line 29: suggest rephrase to “,,,will help reduce disability and deaths from neonatal jaundice.”

Authors response

Page 2 line 67-68: The abstract aspect of the manuscript has been revised to include the reviewer’s suggestion. 

INTRODUCTION

Reviewer’s comment

Page 3 line 14: it will be useful to also capture global targets (SDG)

Authors response

Page 5 line 104-107: The introduction section of the manuscript has been revised to include the global targets of reducing neonatal deaths

Reviewer’s comment

Page 3 lines 9-11. Although neonatal jaundice affects a number of babies, many of them recover. A few may die from kernicterus or bilirubin encephalopathy however underlying causes such as neonatal sepsis are more often the cause of neonatal death in babies with jaundice. Nonetheless neonatal jaundice is a major cause of disability i.e. cerebral palsy and emphasis of this role is lacking. Thus need to rephrase to capture this. 

Authors response

Page3 line 92-96: The authors agreed that the above suggestion is relevant in the area of neonatal jaundice as it will reveal the deleterious effects of jaundice to the reader. Given this, the introduction section of the manuscript has been revised to capture this information. 

Reviewer’s comment

Page 4 line 9: should read “...as a means of treatment”

Authors response

Page 5line 121-122: The manuscript has been revised to include this suggestion.

Reviewer’s comment

Page 4 line 10 change “generations back” to “previous generations”

Authors response

Page 5 line 126: The authors changed “generations back” to “previous generations”.

Reviewer’s comment

Page 4 line18 suggest rephrasing to read...serve as useful information

Authors response

Page 5 line 134: The manuscript was revised and the phrase “serve as a useful information” was changed to “serve as useful information”.

METHODS

Reviewer’s comment

Page 5 ethical approval: were the benefits and risks also explained to them?

Authors response

Page 6 line 149-154: The authors at the point of data collection explained the ethical issues including benefits and risks to the caregivers. However, in the final write up of the manuscript, the benefits and risks aspects were not included in the ethical consideration as this was an oversight. We acknowledge the importance of benefits and risks to the participants. Because of the reviewer’s comments, the manuscript has been revised to include the benefits and risks.

Reviewer’s comment 

Study area: Information on the delivery rate, presence of a newborn unit, its capacity, admission and neonatal mortality rates will be useful.

Authors response

Page 6-7 line 160-166: The manuscript was revised to include the above comment. 

Reviewer’s comment

Page 5 line 17: ...the health facility serves as a major facility for neighbouring Togo residents – needs to be put in the right perspective as unintentional. 

Authors response

Page 6: The authors revised the manuscript and have decided to remove this portion “the health facility serves as a major facility for neighbouring Togo residents” from the study area.

Reviewer’s comment

Page 6 line 17 bed capacity” remove approximately- be specific

Authors response

Page 6 line 160: The authors removed “approximately” from the study area section of the manuscript.

Reviewer’s comment

Page 6 study design – reads better without the abbreviations. i.e. if they are written in full -. KAP and NJ

Authors response

Page 7 line 172: The authors have revised the manuscript and KAP and NJ have been written in full as “knowledge, attitude and practice” and “neonatal jaundice”.

Reviewer’s comment

Page 6 line 20 - 21 Inclusion criteria- need to include the recruitment period

Authors response

Page 8 line 197: The manuscript was revised in the inclusion and exclusion section to include the recruitment period of the caregivers.

Reviewer’s comment

Page 7 line 1: rephrase to “...not the parents of the baby” rather than “...not the real father or mother”

Authors response

Page 8 line 196: In the inclusion and exclusion section, the phrase “not the real father or mother” was revised to “not the parents of the babies”.

Reviewer’s comment

Page 7 line 7 - no mention of northern tribes – they form a significant proportion of the Ghanaian population, were they few? 

Authors response

From the 2010 national housing and population census, Ewes constituted the majority (73.8%) of the ethnic groups in Volta region, followed by Gurma (11.3%) and the Guan (8.1%). Specifically, the largest ethnic group in the Ho municipality where the study was conducted was Ewes (91.1%) followed by Akan (2.0) and the northern tribes (0.5%). However, the authors did not have any empirical data regarding the ethic stratification in the volta region and especially Ho at the time of designing the data collection instrument. Therefore, based on the statistics above, the authors decided to put all other tribes aside those mentioned in the study as “others ethnicity” which includes the northern tribes as contained on page 8 (study variables) and 12 (demographic characteristics).

Ghana Statistical Service (2013). 2010 Population and Housing Census. Regional Analytical Report. GSS, Volta Region

Reviewer’s comment

Page 7 line 12: Adequacy of antenatal visits is now pegged at 8 or more visits but 4 or more can still be used as information on this is more readily available for comparison. Need to mention

Authors response

Page 9 line 208-210: The authors are grateful for this revelation on the adequacy of Antenatal Visit being pegged at 8 or more visits. We have therefore mentioned this in our manuscript but have maintained the 4 visits or more since the data was taken based on this. 

Reviewer’s comment

Page 7 line 15 -17. Add - these are captured in more details in the results section.

Authors response

Page 9 line 215-216: The study variables section of the manuscript was revised to include the phrase “these are captured in more details in the results section” of the manuscript.

Reviewer’s comment

Page 8 line 17: “...further explanation” seems a more appropriate to understanding 

Authors response

Page 10 line 243: The data collection section was revised where “further understanding” was replaced with “further explanation”.

Reviewer’s comment

Page 9 line 4 comment of significance of these coefficients. 

Authors response

Page 11 line 253-260: The manuscript was revised to include the significance of the coefficient values. 

RESULTS

Reviewer’s comment 

Page 10. Table 1 title: choose caregivers or participants 

Authors response

The manuscript was revised replacing “participants” with “caregivers”. 

Reviewer’s comment 

Page 13: frequent ANC attendance alone will not prevent jaundice unless it provides education on prevention and recognition. Sentence should reflect this.

Authors response

Page 16: The authors largely agreed to the above comment and have therefore revised this item under the caregivers' attitude regarding neonatal jaundice.

Reviewer’s comment 

Page 13 Lines 4 and 5 only a few identified traditional practices as being wrong should be reflected in the discussion and conclusion

Authors response

Page 26 line 454-459: The manuscript was revised in the discussion and conclusion sections to include the reviewer’s suggestion on.

Reviewer’s comment

Page 10. Table 1 an unusually large number of caregivers with tertiary education needs an explanation and additional information on catchment area for hospital or was there a selection before the sampling method was applied 

Authors response

The Ho Teaching Hospital is located in the Ho municipality. The municipality is predominantly an urban area. Data from the Ghana Statistical Service (2013) showed that about 53.9% and 46.1% of women and men in the municipality are literate. Also, 30.8% of the women and 34.0% of men in the municipality can speak English and other languages. 

The increased number of caregivers with tertiary education in the current study could have been a coincidence since data collection was mainly through systematic random sampling. As a teaching hospital, many elite families who are highly educated would prefer to seek care there to receive the best care and this could have increased the number of caregivers with tertiary education. The authors employed the sampling strategy in selecting the caregivers, therefore, there was no selection of participants before applying the sampling strategy.

Reviewer’s comment 

Page 13 Attitude questions seem like knowledge questions.

Authors response

Page 17: The attitude questions were revised in the main manuscript.

Reviewer’s comment 

Page 13 last but one sentence spelling of colour –color

Authors response

The word “color” was revised to “colour” in the main manuscript.

Reviewer’s comment 

Page 17 last line & Table 7 suggests - factors associated rather than determinants. Having one child was associated p-0.009 (with) ones practices, but not explained or discussed

Authors response

The authors have revised the results section of the manuscript to change “determinants” to “factors associated with” from page 17-22. Also, in the description section of the multivariate analysis of caregivers’ attitudes regarding neonatal jaundice, caregivers who had one child were 58% less likely to demonstrate a positive attitude of neonatal jaundice. This is captured on page 19 line 352-354 of the revised manuscript.

Reviewer’s comment 

Conclusion poor knowledge regarding traditional treatments not mentioned

Authors response

Page 27 line 477-480: The conclusion section of the has been revised to include this comment.

DISCUSSIONS

Reviewer’s comment 

Page 20 line 22: Is it possible that the reason for the disparity between your finding and Amegan-Aho et al‘s is because there were several caregivers with tertiary education in this study?

Authors response

In our study, the majority of caregivers had tertiary education (45.0%) than Amegan-Aho et al (2019) but demonstrated poor knowledge (8.9%) compared to 76% good knowledge in Amegan-Aho et al with only 5.1% tertiary education. 

Ideally, one would have thought that a high level of tertiary education would be associated with a good knowledge in health-related issues as revealed by Greenaway et al (2012) and low level of education will lead to poor health-seeking behaviours including neonatal jaundice (Ogunlesi & Abdul, 2015). As revealed in our study on page 23 line 404-407, there was no association between caregivers’ educational status with knowledge on neonatal jaundice.

Greenaway, E. S., Leon, J., & Baker, D. P. (2012). Understanding the association between maternal education and use of health services in Ghana: exploring the role of health knowledge. Journal of biosocial science, 44(6), 733.

Ogunlesi, T. A., & Abdul, A. R. (2015). Maternal knowledge and care. Seeking behaviors for newborn jaundice in Sagamu, Southwest Nigeria. Nigerian journal of clinical practice, 18(1), 33-40.

Reviewer’s comment 

Page 21 line 6 suggest “...though in the minority...”

Authors response

Page 24 line 415: The discussion section was revised to capture “though in the minority”.

Reviewer’s comment 

Page 21 line 8 suggest “... poor knowledge of (omit “a”)”

Authors response

Page 24 line 417: The manuscript was revised in the discussion section to remove “a” and include “of”

Reviewer’s comment 

Page 21 line 14 suggest “... knowledge of... rather than “in”.

Authors response

Page 24 line 417: The manuscript was revised in the discussion section to replace “in” with “of”.

Reviewer’s comment 

Page 21 last line: what is meant by “positive” knowledge?

Authors response

Page 24 line 426: The intention was to indicate good knowledge and not positive knowledge. This has been revised as “good knowledge” and not “positive knowledge”.

Reviewer’s comment 

Page 22 last but 3 line suggest removal of “will” to read who “offer”

Authors response

Page 26 line 463: The word “will” was removed from the sentence.

DISCRETIONARY CORRECTIONS

Reviewer’s comment 

Page 4 line 7 suggest revising “where” to read “because”. 

Authors response

Page 5 line 123: Revision was made in which “where” was changed to “because”.

Reviewer’s comment 

Page 8 line 11: questions asked in the questionnaire –suggest rephrase

Authors response

Questions in the questionnaire has been revised from 15-17.

Reviewer’s comment 

Page 20 Line 12: suggest revising “...where majority to “in which majority” 

Authors response

Page 23 line 396-397: Revision was made in the discussion section in which “where majority” was changed to “in which majority”.

Reviewer’s comment 

Page 21 line 17 suggest removal of “pre” before suggested

Authors response

Page 25 line 429: Revision was made and “pre” was removed.

---

## [Decision Letter · Decision Letter 1]

12 Mar 2021

PONE-D-20-19725R1

Manuscript title: Knowledge, Attitudes and Practices Regarding Neonatal Jaundice Among Caregivers in a Tertiary Health Facility in Ghana.

PLOS ONE

Dear Dr. Salia,

Thank you for submitting your manuscript to PLOS ONE. After careful consideration, we feel that it has merit but does not fully meet PLOS ONE’s publication criteria as it currently stands. Therefore, we invite you to submit a revised version of the manuscript that addresses the points raised during the review process.

We look forward to receiving your revised manuscript.

Kind regards,

Bolajoko O. Olusanya, MBBS, FMCPaed, FRCPCH, PhD

Academic Editor

PLOS ONE

Journal Requirements:

Reviewers' comments:

Reviewer's Responses to Questions

**Comments to the Author**

1. If the authors have adequately addressed your comments raised in a previous round of review and you feel that this manuscript is now acceptable for publication, you may indicate that here to bypass the “Comments to the Author” section, enter your conflict of interest statement in the “Confidential to Editor” section, and submit your "Accept" recommendation.

Reviewer #1: All comments have been addressed

Reviewer #2: (No Response)

2. Is the manuscript technically sound, and do the data support the conclusions?

Reviewer #1: Yes

Reviewer #2: Yes

3. Has the statistical analysis been performed appropriately and rigorously? 

Reviewer #1: Yes

Reviewer #2: I Don't Know

4. Have the authors made all data underlying the findings in their manuscript fully available?

Reviewer #1: Yes

Reviewer #2: (No Response)

5. Is the manuscript presented in an intelligible fashion and written in standard English?

Reviewer #1: Yes

Reviewer #2: Yes

6. Review Comments to the Author

Reviewer #1: The reviewer is satisfied by the responses given to questions raised towards the original MS contents. I believe the article is now in a better shape to be accepted for publication and so recommend same.

Reviewer #2: General Comment.

The manuscripts reads much better and most of my previous comments have been addressed. However, there are still some minor essential corrections and discretionary corrections that need to be addressed. Suggest further English language editing.

Minor Essential Corrections

Abstract

Page 2 line 47: suggest changing “...of neonatal jaundice...” to “...regarding neonatal jaundice...”

Page 2 line 52-53: suggest revision: “...sampling strategy. Quantitative data was collected using a questionnaire and analysed with STATA....

Page 2 line 54: the phrase “...neonatal jaundice knowledge...” does not seem right, kindly revise.

Results

Page 2 line 57 sentence beginning with “More than 50% (54.5%) and (52.5%) of caregivers...” is unclear, kindly revise

Page 2 line 59: suggest revising “....knowledge in...” to “knowledge about...”

Conclusion – needs to be more specific and focused on findings.

Page 2 Line 65: suggest changing “...of jaundice...” to “...about jaundice...”

Introduction

Page 3 line 76-77: “Jaundice ...manifestation of a disease.” kindly add a reference.

Page 3 line 99: reference 12: Reference from Graphic should be changed to a more appropriate source of data on prevalence of neonatal jaundice in Ghana.

Page 3 line 101 suggest changing revealed to reported

Materials and methods

Page 6 line146: the word reinforced is unclear in this context.

Page 6 line 153-154: invasive procedures are not the only sources of risk or discomfort so suggest you remove or revise the reference. The inconvenience of additional waiting to complete the questionnaire, though minimal may be seen as a source of discomfort to some.

Page 6 line 157: study area you have described the neonatal unit however it is still not clear how neonatal services are delivered at the hospital as you have not described the postnatal ward in adequate detail.

Page 7 line 172: remove capitalisation

Page 7 line 175: Change methods to determination to meet journal specification

Page 7 line 184: sentence “Balloting was done...”. needs to be re-sequenced as the skip interval is usually determine before the starting point.

Page 9 line 208 to 2010: suggest revision- Although WHO has changed the minimum recommended antenatal visits from 4 to 8 visits, in this study we studied an attendance of 4 visits.[reference] No need to mention availability of information for comparison..

Page 9 line 218-219 remove capitalisation

Page 10 line 238 remove” For”

Results

Format tables to fit with journals recommendations and also provide information on Total respondents

Table1: Title - change caregiver to caregivers; Characteristics - Education received on neonatal jaundice and Child with a diagnosis of neonatal jaundice reads better (page 14)

Table 2: title- suggest - Caregivers knowledge about neonatal jaundice; Prevention of neonatal jaundice: Neonatal jaundice is a common problem on newborns...so there is no need to prevent it reads better

Table 5 and 6; more information needed on the logistic regression model. What aspects of the knowledge, attitudes, beliefs and practices and were examined?

Page 19 line 354. There is no decimal place in the p-value.

Conclusion: Does not provide a succinct summary of some of the good practices and gaps in knowledge.

Lines 477-480 no need to bring in percentages.

Discretionary corrections

Page 2 line 42: suggest removing word “trends”

Page 8 line 198; suggest removing “thus”

References - Need to examine references again. Inconsistency in formatting noted,; date of accessing online publication omitted in some instances

---------

7. PLOS authors have the option to publish the peer review history of their article (what does this mean?). If published, this will include your full peer review and any attached files.

Reviewer #1: **Yes: **Stephen Oguche

Reviewer #2: No

---

## [Author Response · Author response to Decision Letter 1]

20 Apr 2021

Please see response to reviewers letter at the end of the PDF for review

---

## [Editor Report · Decision Letter 2]

29 Apr 2021

PONE-D-20-19725R2

Manuscript title: Knowledge, Attitudes and Practices Regarding Neonatal Jaundice Among Caregivers in a Tertiary Health Facility in Ghana.

PLOS ONE

Dear Dr. Salia,

Thank you for submitting your revised manuscript to PLOS ONE. This current version is much improved but the key messages are still hazy and imprecise. At this stage I invite you to revise your abstract by addressing the following edits carefully to improve the clarity of your message to our audience. You will then need to amend your results and conclusions in line with the suggested changes in the abstract. 

1. Abstract - Results 

DELETE: ‘*Most* of the caregivers demonstrated poor knowledge (54.5%) and attitude (52.5%) while 58.9% 

had good practices regarding neonatal jaundice’. 

REPLACE WITH:  ‘Less than half of the caregivers demonstrated good knowledge (45.5%) and attitude (47.5%) but 58.9% had good practices regarding neonatal jaundice’.

2. Abstract Conclusion:

DELETE: ‘Most of the caregivers demonstrated poor knowledge and attitude about jaundice while the majority demonstrated good practice. Healthcare professionals need to intensify education regarding jaundice among caregivers in low resource settings throughout antenatal and postnatal periods. Good knowledge may possibly lead to positive attitudes towards jaundice. Overall, improved knowledge and attitude improve maternal health-seeking behaviours which may help to reduce disabilities and deaths from neonatal jaundice’.

REPLACE WITH: 'Less than two thirds of the caregivers demonstrated good practice with limited knowledge and poor attitude. Efforts to promote well informed and improved caregivers’ attitude will advance positive maternal health-seeking behaviour and reduce disabilities and death through early detection and intervention of infants with neonatal jaundice. Public awareness and education about neonatal jaundice especially among caregivers in the private sector should also be intensified'. 

3. Reference 12 is incomplete without a URL.

We look forward to receiving your revised manuscript.

Kind regards,

Bolajoko O. Olusanya, MBBS, FMCPaed, FRCPCH, PhD

Academic Editor

PLOS ONE
---

## [Author Response · Author response to Decision Letter 2]

2 May 2021

Manuscript title: Knowledge, Attitudes and Practices Regarding Neonatal Jaundice Among Caregivers in a Tertiary Health Facility in Ghana

*Solomon Mohammed Salia1, Agani Afaya1,2, Abubakari Wuni3, Martin Ayanoore4, Emmanuel Salia5, Doreen Dzidzor Kporvi1, Peter Adatara1, Robert Kaba Alhassan6, Vida Nyagre Yakong7, Sean Augustine Eduah-Quansah1, Shine Seyram Quarshie1, Eric Kwame Dey1, Dominic Amoah Akolga1

Dear editor, 

We are very happy that you are giving us another chance to shape our manuscript to the required journal’s standard for publication. We would like to thank the reviewers for the thorough review and insightful comments on our manuscript. The comments though minor are very useful and we have responded to them to the best of our knowledge. We acknowledge that the comments have no doubt helped improve the quality of our manuscript.

We have therefore provided further details by showing point-by-point feedback on how each of the comments was addressed. For easy identification of our responses, the comments have been repeated while the Authors’ responses appear in BOLD text in the main manuscript.

Editor’s Comments

Abstract - Results 

DELETE: ‘Most of the caregivers demonstrated poor knowledge (54.5%) and attitude (52.5%) while 58.9% had good practices regarding neonatal jaundice’. 

REPLACE WITH: ‘Less than half of the caregivers demonstrated good knowledge (45.5%) and attitude (47.5%) but 58.9% had good practices regarding neonatal jaundice’.

Authors Response

Page 2 line 60-61: The results section of the abstract was deleted and replaced with the suggested statement by the editor which adds more meaning to the results.

Editor’s Comment

2. Abstract Conclusion:

DELETE: ‘Most of the caregivers demonstrated poor knowledge and attitude about jaundice while the majority demonstrated good practice. Healthcare professionals need to intensify education regarding jaundice among caregivers in low resource settings throughout antenatal and postnatal periods. Good knowledge may possibly lead to positive attitudes towards jaundice. Overall, improved knowledge and attitude improve maternal health-seeking behaviours which may help to reduce disabilities and deaths from neonatal jaundice’.

REPLACE WITH: 'Less than two thirds of the caregivers demonstrated good practice with limited knowledge and poor attitude. Efforts to promote well informed and improved caregivers’ attitude will advance positive maternal health-seeking behaviour and reduce disabilities and death through early detection and intervention of infants with neonatal jaundice. Public awareness and education about neonatal jaundice especially among caregivers in the private sector should also be intensified'. 

Authors Response

Page 2 line 68-73: The authors have deleted the conclusion section of the abstract and have replaced it with the editor’s suggested phrase. This has made the understanding of the conclusion clearer.

Editor’s Comment

3. Reference 12 is incomplete without a URL.

Authors Response

Reference to the second revision, the reviewer requested that we change the graphic Online source to a more appropriate reference. However, the authors have searched and have realized that the information used by graphic online was from the Ghana Health Service (GHS), but this information from the GHS was not published as at the time the authors used the information, and therefore, did not have a URL. 

Page 31 line 610-612: The authors have revised the reference and have added the URL

---

## [Editor Report · Decision Letter 3]

5 May 2021

Manuscript title: Knowledge, Attitudes and Practices Regarding Neonatal Jaundice Among Caregivers in a Tertiary Health Facility in Ghana.

PONE-D-20-19725R3

Dear Dr. Salia,

We’re pleased to inform you that your manuscript has been judged scientifically suitable for publication and will be formally accepted for publication once it meets all outstanding technical requirements.

Kind regards,

Bolajoko O. Olusanya, MBBS, FMCPaed, FRCPCH, PhD

Academic Editor

PLOS ONE
---

## [Editor Report · Acceptance letter]

26 May 2021

PONE-D-20-19725R3 

Knowledge, Attitudes and Practices Regarding Neonatal Jaundice Among Caregivers in a Tertiary Health Facility in Ghana. 

Dear Dr. Salia:

I'm pleased to inform you that your manuscript has been deemed suitable for publication in PLOS ONE. Congratulations! Your manuscript is now with our production department. 

Kind regards, 

on behalf of

Dr. Bolajoko O. Olusanya 

Academic Editor

PLOS ONE